# Effect of Different Sterilization Methods on the Microbial and Physicochemical Changes in *Prunus mume* Juice during Storage

**DOI:** 10.3390/molecules27041197

**Published:** 2022-02-10

**Authors:** Yuan Ma, Yingping Xu, Yuanyuan Chen, Ailian Meng, Ping Liu, Kunyue Ye, Anqi Yuan

**Affiliations:** Key Laboratory of Grain and oil Processing and Food Safety of Sichuan Province, College of Food and Bioengineering, Xihua University, Chengdu 610039, China; yingpingshou@163.com (Y.X.); chenyuanyuan0714@163.com (Y.C.); mngailian2021@163.com (A.M.); Liuping20210414@163.com (P.L.); y19140556253@163.com (K.Y.); yuananqi980620@163.com (A.Y.)

**Keywords:** *Prunus mume* juice, sterilization methods, physicochemical property, microorganisms, storage

## Abstract

This study evaluated the pasteurization (P), ozone (O_3_), ultrasonic (US), and high-hydrostatic-pressure (HHP) sterilization approaches for processing of *Prunus mume* regarding browning factors and microorganisms, compared with non-sterilization (control check, CK) treatment. The microorganisms (total bacterial count and fungi and yeast count) in the juice were identified after different sterilization techniques, while the quality parameter changes (degree of browning, color measurements, total phenolic content, reducing sugar, ascorbic acid, 5-hydroxymethyl furaldehyde (5-HMF), amino acid nitrogen, total soluble solids (TSS), pH value) were investigated. The results indicate that P and HHP treatment reduced non-enzymatic browning while substantially impacting the color measurements, TSS, and pH, while the sterilization effect was remarkable, with a rate exceeding 90%. Furthermore, the *Prunus mume* juices treated with P and HHP sterilization were used as the objects, and the CK group was used as the control group. They were placed at 4 °C, 25 °C and 37 °C, respectively, and stored in dark for 15 d. Sampling and determination were carried out on 0, 3, 6, 9, 12, and 15 d, respectively. M-&-Y (molds and yeasts) were not detected in the late storage period, and no obvious microbial growth was observed during storage, indicating that P and HHP treatments could ensure the microbial safety of *Prunus mume* juice. P- and HHP- treated *Prunus mume* juice has better quality and low temperature storage is beneficial for maintaining the quality of *Prunus mume* juice. Therefore, P treatment or HHP treatment combined with low temperature storage could achieve a more ideal storage effect. Overall, this study conclusively established that P and HHP methods were suitable for sterilizing *Prunus mume* juice. These techniques minimally affected overall product quality while better maintaining the quality parameters than the untreated juice samples and those exposed to O_3_ and US treatment.

## 1. Introduction

*Prunus mume Sieb. et Zucc.*, popularly known as Chinese plum, is one of the most important seasonal fruits in southern China. *Prunus mume* is a uniquely flavored, highly acidic yellow or green fruit [1] with a sugar-to-acid ratio of 0.2 [2]. Known as Fructus mume in its unripe state, it is used in traditional medicinal food. Since *Prunus mume* is rich in organic acids, amino acids, vitamins, flavonoids, and polyphenols and presents antibacterial [3], antiallergic [4], and antioxidant activity [5], it is an excellent natural health food. The water content in fresh *Prunus mume* generally exceeds 88%, which is suitable for juice, however, its acidity renders it unacceptable to consumers. Therefore, *Prunus mume* juice is mixed with other materials to produce compound beverages after juice extraction, which forms the markets demand for *Prunus mume* juice.

Processing and storage lead to substantial *Prunus mume* juice browning [6], which can be classified into enzymatic and non-enzymatic browning. Enzyme-driven browning mostly occurs during the crushing, juicing, and filtration stages, while the oxidation that occurs during sterilization and the subsequent storage process is primarily ascribed to non-enzymatic browning. Thermal sterilization, traditionally used for fruit and vegetable juice, is not suitable for nutrient preservation, quickly leading to browning and a decline in product quality [3]. Contrarily, non-thermal sterilization is highly successful in retaining the original nutritional composition and quality of juice and mainly includes ozone (O_3_) [7,8], ultrasonic (US) [9], high-hydrostatic-pressure (HHP) [10], and high-voltage pulsed electric field sterilization [11]. Such processes minimize the negative thermal effects on food nutritional and quality parameters [12], while ensuring safety from a microbiological point of view. The application efficiency of ozone in fruit juices has been studied, mainly in apple and orange juices [13] focusing on quality and safety characteristics. You [14] showed that compared with the traditional thermal sterilization treatment, HHP sterilization treatment can better reduce the damage to anthocyanin and improve the quality of mulberry juice. However, no studies are available regarding its application to *Prunus mume* juice. Therefore, this work compared the effect of pasteurization (P), as well as O_3_, US, and HHP sterilization, on the physicochemical properties (degree of browning, color measurements, total phenolic content, reducing sugar, ascorbic acid, 5-hydroxymethylfurfural, amino acid nitrogen, total soluble solids (TSS), pH value) and microorganisms total number of the bacterial colony and aspergillus and yeast (APC and M&Y) of *Prunus mume* juice. Furthermore, the changes in physicochemical indexes and microorganism of *Prunus mume* juice treated by P and HHP treatments during storage were further observed to identify the most suitable technique for *Prunus mume* juice application.

## 2. Results and Discussion

### 2.1. The Effect of Different Sterilization Methods on the Browning Degree and Color Measurements of Prunus mume Juice

The degree of browning denotes a reference index to measure the browning of juice during the sterilization treatment process. Figure 1 shows that the browning degree of P-, O_3_-, US-, and HHP-treated *Prunus mume* juice decreased significantly compared with non-sterilization treatment (*p* < 0.05). Although oxidation changes were more distinct during P and HHP treatment, no significant differences were evident between the two groups (*p* > 0.05), indicating that both low-temperature P and HHP sterilization had a less substantial impact on the browning degree of *Prunus mume* juice.

Colorimetric analysis of the color characteristics after the different sterilization treatments indicated distinct changes. The impact of P, O_3,_ US, and HHP processing on color is shown in Table 1. The L* value of the P-, O_3_-, and HHP-treated juice exhibited a significant decrease (*p* < 0.05) compared with freshly squeezed juice, while that of the US-treated juice remained unchanged (*p* < 0.05). The a* value of the samples treated with P, US, and HHP increased significantly (*p* < 0.05), while that of the O_3_-treated juice exhibited a substantial decline (*p* < 0.05) compared with freshly squeezed juice. A considerable increase in the b* value was evident in all the treated juice samples compared with untreated juice. The ΔE value indicated significant color differences (*p* < 0.05) between the untreated and treated samples, as shown in Table 1. The ΔE values between the sterilization treatment groups and the non-sterilization treatment group were P > O_3_ > US > HHP > Non-sterilization (*p* < 0.05) in descending order. This showed that P treatment had the most significant impact on the *Prunus mume* juice color, which could be attributed to the fact that the thermal effect during heat treatment accelerated the browning of *Prunus mume* juice [14,15]. Jabber [16] found that thermal treatment reduces the activity of enzymes, which in turn inhibits browning caused by enzymes and hence improves the color of the product. The effect of O_3_ treatment on color may be ascribed to the attack of coupling double bonds by O_3_ molecules and hydroxyl radicals, resulting in the partial oxidation of organic acids, aldehydes, and ketones [17]. HHP treatment had an effect on the *Prunus mume* juice color, but it was better at maintaining the original color compared to other sterilization treatments. Justyna [18] indicated that HHP-treated carrot juices were less red and yellow compared with fresh carrot juice. Elizabeth [19] indicated that there was a significant (*p* < 0.05) effect of pressure on color characteristics of pomegranate juice with increased pressure and that these decreases in a* and L* values can be attributed to the degradation or polymerization of anthocyanins and indicate a fading of the typical red color of pomegranate juice. Therefore, HHP can better maintain the color of *Prunus mume* juice in terms of the changes in browning degree and color.

### 2.2. The Effect of Different Sterilization Methods on the Total Phenols and Reducing Sugar in the Prunus mume Juice

The impact of different sterilization treatments on the total phenols is shown in Figure 2a. The initial phenolic content in the control sample (untreated) was 0.0647 mg GAE/mL while decreasing significantly (*p* < 0.05) in the P-, O_3_-, and HHP-treated juice. No significant differences were evident between the US and HHP-treated juice. The total phenolic content in the *Prunus mume* juice decreased after P treatment, which could be attributed to the thermal instability of phenolic compounds. High temperatures accelerated the degradation of some phenolic compounds [19]. The total phenolic content loss was the highest in the O_3_-treated juice at 0.0551 mg GAE/mL, which could be ascribed to various chemical reactions, including direct chemical reactions between the O_3_ and the target compounds or their intermediates or free radical reactions between the hydroxyl radicals [20], which was consistent with the findings of Tiwari [21,22]. The decreased total phenolic content in *Prunus mume* juice after HHP treatment may be due to an increase in the extractability of phenolic compounds or non-enzymatic oxidative degradation reaction in samples after high-pressure treatment [23]. The effect of US treatment on the total phenolic content in *Prunus mume* juice was relatively mild. Saeeduddin [24] reported similar research results, indicating that US sterilization could better maintain polyphenols in fruits and vegetables. Moreover, Bhat [25] found that the total phenolic content increased significantly after US treatment. 

The impact of the different sterilization treatments on reducing sugar is shown in Figure 2b. The reducing sugar content in the US-treated *Prunus mume* juice exhibited a significant decrease (*p* < 0.05) at 0.618 mg/mL, which could be attributed to the thermal, cavitation, and mechanical impact [26]. The reducing sugar content increased substantially in the *Prunus mume* juice after O_3_ treatment (*p* < 0.05) at 0.666 mg/mL, which could be ascribed to the reaction between O_3_ and polysaccharides, resulting in the glycosidic bond cleavage and hydrolysis to produce reducing monosaccharides [27]. The decrease in reducing sugars by ultrasound and heat treatment may be due to the formation of 5-HMF via sugar dehydration and amino acid interaction at elevated temperatures [28]. The results indicate that HHP treatment could maintain the reducing sugar content in the *Prunus mume* juice since no significant differences (*p* > 0.05) were evident, as was the case with P (*p* > 0.05).

### 2.3. The Effect of Different Sterilization Methods on the 5-Hydroxymethyl Furaldehyde Levels and the Amino Acid Nitrogen Content in the Prunus mume Juice 

The impact of different sterilization treatments on 5-HMF is shown in Figure 3a. The 5-HMF content in the *Prunus mume* juice treated with P, O_3_, and HHP did not exhibit a significant change (*p* > 0.05) but increased considerably (*p* < 0.05) in the US treatment group. The 5-HMF is mainly produced by the Maillard reaction and the dehydration of carbohydrates under acidic conditions. Therefore, on the one hand, this may involve the Maillard reaction between β-lactoglobulin and glucose, galactose, lactose, fructose, ribose, and arabinose in a water system, induced by high-intensity US treatment. On the other hand, this may denote the hydrolysis of reducing sugar under acidic conditions. Zhang [29] found that acidic and high-temperature experimental conditions favor the fructofuranosyl cation pathway, while a higher pH and lower temperature are conducive to HMF formation.

Amino acid nitrogen indicates the amino acid content in food while indirectly reflecting the Maillard reaction [19]. As shown in Figure 3b, no significant differences were evident in the amino acid nitrogen levels in the *Prunus mume* juice after P, US, and HHP treatment (*p* > 0.05), while a substantial decrease (*p* < 0.05) was apparent in the samples treated with O_3_, at a level of 0.0031 mg/mL. The reason for this may be that O_3_ can oxidize amino acid components (cysteine, tryptophan, methionine, and histidine) and glycolipids in cell membrane proteins, decreasing the amino acid content in *Prunus mume* juice. In addition, O_3_ may aggravate the Maillard reaction in the juice, leading to amino acid consumption.

### 2.4. The Effect of Different Sterilization Methods on the Total Soluble Solid and pH of the Prunus mume Juice 

The TSS content in the *Prunus mume* juice treated with P, O_3_, US, and HHP was significantly higher than in the non-sterilization treatment, as shown in Table 2 (*p* < 0.05), suggesting that these treatments were beneficial to TSS dissolution. The reason for the highest TSS content under US treatment may be that ultrasonic treatment accelerated the hydrolysis of reducing sugar under acidic conditions. Furthermore, the juice displayed substantially higher sucrose, fructose, and glucose levels [19]. 

The pH values of the P-treated samples were considerably higher than in the non-sterilization treatment (*p* < 0.05), indicating that the treatment strategies could cause strong physical or chemical reactions in the juice, leading to acid substance formation. However, the pH value of the US- and O_3_- and HHP-treated samples changed slightly. These results are in line with the findings of Abid [30,31,32]. This may be attributed to a higher citric acid level in the form of free acid, as well as the formation of buffer substances with inorganic salts and other substances, preventing significant pH changes. Additionally, Jaramillo-Sánchez [33] found that no significant difference was observed for pear juice treated with ozone. Lower ozone concentration has no effect on pH and titratable acidity of fruit juices [34].

### 2.5. The Effect of Different Sterilization Methods on the Microbial Counts in the Prunus mume Juice 

The APC and M&Y content in the *Prunus mume* juice treated using different sterilization methods are listed in Table 3, showing distinct variations in their impact on the juice samples. Compared with the CK group, the P, O_3_, US, and HHP sterilization methods had bactericidal effect, and no M&Y content was detected. P exhibited the best sterilization effect, while APC and M&Y were not detected after treatment. Furthermore, the bactericidal effect of different sterilization methods on APC content was P, HHP, US, O_3_. Therefore, P, O_3_, US, and HHP sterilization decreased the microbial indexes of the *Prunus mume* juice. Shah [17] demonstrated that ozone treatment reduces the number of microorganisms in pummelo fruit juice. Khandpur [35] evaluated ultrasound-based sterilization approaches showed an obvious inhibitory effect on microorganisms.

### 2.6. Changes in Physicochemical Properties and Microorganisms during Storage with Pasteurization and High-Hydrostatic-Pressure Sterilization

#### 2.6.1. Changes in Browning Degree and Color Measurement of *Prunus mume* Juice during Storage with Pasteurization and High-Hydrostatic-Pressure Sterilization

The *Prunus mume* juices of P, HHP, and CK sterilization groups were stored at different temperatures (4 °C, 25 °C, 37 °C), and the browning changes during storage are shown in Figure 4a. After 15 days storage, compared with untreated juice, the browning degree of HHP- and P-treated juice changed little, and the browning degree of HHP was more stable. At 4 °C storage temperature, the browning degree of *Prunus mume* juice treated with P and HHP fluctuated slightly, but there was no significant difference (*p* > 0.05). The browning degree of *Prunus mume* juice in CK group increased from 4.55 ± 0.11 to 5.16 ± 0.03. At 25 °C storage temperature, the browning degree of *Prunus mume* juice treated with P increased from 3.85 ± 0.06 to 5.14 ± 0.06, the browning degree of juice treated with HHP increased from 3.85 ± 0.06 to 4.49 ± 0.10, and the browning degree of control CK group increased from 4.55 ± 0.11 to 6.28 ± 0.14. At 37 °C storage temperature, the browning degree of *Prunus mume* juice treated with P increased from 3.85 ± 0.06 to 5.93 ± 0.07, the browning degree of juice treated with HHP increased from 3.85 ± 0.06 to 5.30 ± 0.11, and the browning degree of control CK group increased from 4.55 ± 0.11 to 6.42 ± 0.12. It can be seen that with the extension of storage time, different sterilization methods can increase the browning degree of *Prunus mume* juice to varying degrees. The lower the storage temperature, the lower the browning degree of *Prunus mume* juice, and the better the stability. Compared with the control group and pasteurization treatment, the color of *Prunus mume* juice after HHP sterilization was better and the browning degree was smaller.

Color change is an important indicator for measuring the change of sensory quality of fruit and vegetable juice during storage. The juices of P, HHP, and CK groups were stored at different temperatures, and the changes in ΔE value during storage are shown in Figure 4b. After 15 days of storage, the storage temperature had a great influence on the color change of P and HHP treated *Prunus mume* juice. Under the storage condition of 4 °C, the ΔE value of *Prunus mume* juice did not change significantly (*p* < 0.05), but with the increase in storage temperature, the ΔE value of *Prunus mume* juice obtained by the two sterilization treatments changed significantly, reaching 3.98 ± 0.01 and 3.67 ± 0.05 at 37 °C, respectively. The same law can be found in the control group, and the color change was mainly caused by browning reaction, which was consistent with the change trend of browning degree of *Prunus mume* juice during storage. It can be seen that at 4 °C, 25 °C and 37 °C, HHP treatment had the best effect on maintaining the color of *Prunus mume* juice, followed by P treatment, and the color of *Prunus mume* juice in the CK group was the most unstable. This is because compared with heat treatment, HHP treatment does not destroy the covalent bond of molecules. At this time, small molecular substances have strong stability and can better maintain the sensory characteristics such as nutrients and pigments in food [36,37]. Wang [38] and Liu [39] also reported similar research results.

#### 2.6.2. Changes in Total Phenol Content and Reducing Sugar of *Prunus mume* Juice during Storage with Pasteurization and High-Hydrostatic-Pressure Sterilization

The *Prunus mume* juices of P, HHP, and CK groups were stored at different temperatures, and the changes in total phenol content during storage are shown in Figure 5a. After 15 days of storage, at 4 °C storage temperature, the total phenol content of *Prunus mume* juice treated with P and HHP had no significant difference (*p* < 0.05). The total phenol content of control CK group decreased from 0.0647 mg GAE/mL to 0.0591 mg GAE/mL. At 25 °C storage temperature, the total phenol content of *Prunus mume* juice treated with P decreased from 0.0608 mg GAE/mL to 0.0559 mg GAE/mL, the total phenol content of *Prunus mume* juice treated with HHP decreased from 0.0632 mg GAE/mL to 0.0568 mg GAE/mL, and the total phenol content of control CK group decreased from 0.0647 mg GAE/mL to 0.0562 mg GAE/mL. At 37 °C storage temperature, the total phenol content of *Prunus mume* juice treated with P decreased from 0.0608 mg GAE/mL to 0.0514 mg GAE/mL, the total phenol content of *Prunus mume* juice treated with HHP decreased from 0.0632 mg GAE/mL to 0.0558 mg GAE/mL, and the total phenol content of control CK group decreased from 0.0647 mg GAE/mL to 0.0518 mg GAE/mL.

Therefore, with the extension of storage time, different sterilization methods can reduce the total phenol content in *Prunus mume* juice to varying degrees. The higher the storage temperature, the more unstable the total phenol in *Prunus mume* juice. At the same storage temperature, compared with the control group and pasteurization treatment, the total phenols of *Prunus mume* juice after HHP treatment remained better, which may be that the low temperature of HHP treatment delayed the oxidative degradation of phenolic substances by oxygen free radicals formed by dissolved oxygen decomposition in the sample [40,41].

The *Prunus mume* juices of P, HHP and CK groups were stored at different temperatures, and the change in reducing sugar content of *Prunus mume* juice during storage is shown in Figure 5b. After 15 days of storage, at 4 °C, 25 °C and 37 °C, compared with 0 d, the reducing sugar content of *Prunus mume* juice in the three groups increased to varying degrees and finally gradually tended to be flat or decreased. This is contrary to the browning degree and ΔE value of *Prunus mume* juice, which may be because the non-enzymatic browning of juice occurs slowly during storage and the reducing sugar is continuously consumed. However, during the browning process, non-reducing sugars such as sucrose in *Prunus mume* juice will be hydrolyzed under acidic conditions to produce reducing sugars such as glucose and fructose, and the increase in storage temperature will accelerate the hydrolysis reaction to ensure the continuous non-enzymatic browning [42]. Therefore, when the hydrolysis rate was far greater than the consumption rate of reducing sugar in browning reaction, the reducing sugar in *Prunus mume* juice increased. Then, with the non-enzymatic browning reaction, the content of reducing sugar began to stabilize or even decrease. 

#### 2.6.3. Changes in the 5-Hydroxymethyl Furaldehyde Levels and the Amino Acid Nitrogen Content of *Prunus mume* Juice during Storage with Pasteurization and High-Hydrostatic-Pressure Sterilization

5-HMF is an intermediate product of Maillard reaction, which not only leads to the deterioration of color and flavor of fruit and vegetable juice, but also affects its edible safety [42]. The *Prunus mume* juices of P, HHP, and CK groups were stored at different temperatures, and the changes in 5-HMF content during storage are shown in Figure 6a. In the early stage of storage, the content of 5-HMF in the three groups of *Prunus mume* juice was low, and with the extension of storage time, it showed an indigenous upward trend. The 5-HMF of CK group was the fastest, and the increment was also the largest. It can be seen that P and HHP treatments can effectively inhibit the formation of 5-HMF, thereby inhibiting the browning reaction of *Prunus mume* juice. The increase in 5-HMF content is relatively slow at 4 °C storage temperature. When the storage temperature is 25 °C and 37 °C, the increase rate of 5-HMF content is larger. Therefore, the increase in storage temperature will accelerate the formation of 5-HMF in *Prunus mume* juice. During the storage process, the 5-HMF content fluctuated irregularly, which was due to the fact that when the accumulation of 5-HMF reached a certain level, it would participate in the Maillard reaction and polymerize to produce dark substances, resulting in the deepening of the color of *Prunus mume* juice. This process would consume part of 5-HMF. When the consumption rate was less than the generation rate, the content of 5-HMF decreased. 

Maillard reaction is a complex reaction with amino acids and reducing sugars as substrates, and the content of amino acid nitrogen can directly reflect the content of amino acids, so the content of amino acid nitrogen can indirectly reflect the Maillard reaction. The *Prunus mume* juices of P, HHP, and CK groups were stored at different temperatures, and the changes in amino acid nitrogen content during storage are shown in Figure 6b. After 15 days of storage, there was no significant difference in amino acid nitrogen content between P- and HHP-treated *Prunus mume* juice at 4 °C storage temperature (*p* > 0.05). The amino acid nitrogen content of *Prunus mume* juice in the control CK group increased slightly, from 0.0036 mg GAE/mL to 0.0041 mg GAE/mL. At 25 °C and 37 °C, the loss of amino acid nitrogen may be due to Maillard reaction with reducing sugar during storage. The content of amino acid nitrogen in *Prunus mume* juice treated with HHP increased first and then decreased. Ultrahigh pressure would destroy the hydrogen bond of protein molecules, and some polar groups were dissociated, so that the molecules on the surface of protein molecules had the same charge, which promoted the separation of the binding substance and protein, thereby increasing the solubility of protein, dissociating to produce small molecular substances such as amino acids, and increasing its content [43]. Then, with the Maillard reaction, amino acids were consumed, and the hydrolysis rate of protein was less than the consumed rate, resulting in a gradual decrease in the content. Because temperature can also affect the hydrolysis of protein, and thus the P group and CK group of *Prunus mume* juice stored at 25 °C and 37 °C, there is the same trend of amino acid nitrogen content being increased at first and then decreased.

#### 2.6.4. Changes in Total Soluble Solid of *Prunus mume* Juice during Storage with Pasteurization and High-Hydrostatic-Pressure Sterilization

TSS refers to the general term for all compounds dissolved in water in liquid or liquid foods, including sugar, acid, vitamins, minerals and so on [44]. The *Prunus mume* juices of P, HHP, and CK groups were stored at different temperatures, and the changes of TSS content during storage are shown in Figure 7. Although the TSS content in the three groups showed a downward trend with the extension of storage time, the retention rate was above 90.00%, and there was no significant difference between groups (*p* > 0.05). Therefore, in general, TSS was relatively stable during the storage of *Prunus mume* juice. 

The pH value of juice can reflect the degree of deterioration of *Prunus mume* juice quality [45]. The *Prunus mume* juices of P, HHP, and CK groups were stored at different temperatures. The pH changes in *Prunus mume* juice during storage are shown in Table 4. There was no significant difference in pH among the three groups of *Prunus mume* juice stored at 4 °C, 25 °C and 37 °C (*p* > 0.05). This may be because the juice contains more citric acid in the form of free acid, and it contains inorganic salts and other substances to form a buffer, so the pH value did not change significantly. 

#### 2.6.5. Changes in Microbial counts of *Prunus mume* Juice during Storage with Pasteurization and High-Hydrostatic-Pressure Sterilization

The *Prunus mume* juices of P, HHP and CK groups were stored at different temperatures. The changes in APC and M & Y during storage are shown in Table 5. The table shows that different sterilization methods have different sterilization effects on *Prunus mume* juice, that the pasteurization treatment had the best sterilization effect on *Prunus mume* juice, and that APC and M-&-Y were not detected within 15 days after storage. Although HHP treatment failed to completely kill APC, it did not increase during the later storage period. It may be that HHP caused damage to the cells of microorganisms, making them unable to be repaired during storage, thereby being inactivated [46].

## 3. Materials and Methods

### 3.1. Juice Preparation 

The *Prunus mume* was obtained from the Tianqing *Prunus mume* Planting Cooperative (Dazhou, Sichuan), from which fruits of uniform size, color, and maturity were selected and washed. The *Prunus mume* kernels were removed. One volume of the pulp was mixed with three volumes of water and squeezed using a juice extractor (JYL-C91T, Joyoung Co., Ltd., Zhejiang, China). Next, 0.06% of a pectinase (Yuan Ye Biotechnology Co., Ltd., Shanghai, China) was added for enzymatic hydrolysis at 34.6 °C for 1.4 h. A mixture of 0.26% (*m*/*v*) of citric acid, 0.26% (*m*/*v*) of ascorbic acid, and 0.20% (*m*/*v*) of calcium chloride (Cologne Chemicals Co., Ltd, Chengdu, Sichuan, China) was prepared and added to the clarified juice to prevent juice browning. Then, the clarified juice was filtered using a muslin cloth prior to treatment.

### 3.2. Pasteurization Sterilization

For the thermal P treatment, the parameter according to methodology previously reported by Marsellés-Fontanet [47] was used with modifications. The juice samples were packed in low density polyethylene (LDPE) packets (30 mL), sealed using a film sealing machine (SF-150, Afanlao Machinery Co., Ltd., Shanghai, China), and pasteurized in a bain-marie (HH-S4, Jintan Medical Instrument Factory, Jiangsu, China) at 80 °C for 10 min.

### 3.3. Ozone Sterilization

The O_3_ treatment was performed using a method described by Fundo [34] with some modifications. Oxygen was passed through a corona discharge generator (YS-MJCB-S17, Hangzhou Yishi Technology Co., Ltd., Zhejiang, China) to produce O_3_ at 5 g/h. Then, 120 mL of *Prunus mume* juice was subjected to constant stirring (magnetic agitation) to ensure equal O_3_ distribution throughout the sample. The iodometric titration method [48] was used to determine the O_3_ concentration of 7.0 ± 2 mg/m^3^. The sample was prepared after O_3_ treatment for 45 min. The treatments were performed in triplicate.

### 3.4. Ultrasonic Sterilization

The US treatment was performed using a method described by Elżbieta [49] with some modifications. Directly after the fresh juice extraction, sonication was performed at 40 W and 200 W for 15 min using a cleaning bath (Scientz-IID, Ningbo Xinzhi Biotechnology Co., Ltd., Zhejiang, China). The process consisted of a 2 s turn-on and 2 s turn-off cycle. All experiments were conducted in triplicate.

### 3.5. High-Hydrostatic-Pressure Sterilization

The pressure experiments were performed in a laboratory-scale high-pressure processor (HHP-600, Baotou Kefa High-Pressure Technology Co., Ltd., Inner Mongolia, China). The HHP sterilization treatment referenced Elizabeth [19] with modification. The samples were placed in 100 mL bottles and enclosed in the pressure vessel already equilibrated at 4 °C and 500 MPa. The vessel was then pressurized and maintained for 10 min, followed by decompression.

### 3.6. Physicochemical Analysis

#### 3.6.1. Determining the Degree of Browning

The browning degree was measured using the spectrophotometric method described by Zhou [50]. Distilled water was used as a control. A volume of juice (1.5 mL) was mixed with an equal quantity of distilled water at 25 °C and left to stand for 5 min, after which the absorbance was measured at 410 nm using a UV-Vis Spectrophotometer (UV2400, Sunny Hengping Scientific Instrument Co., Ltd., Shanghai, China). The results are expressed as a ten-fold absorbance value.

#### 3.6.2. Color Measurements

The color measurements were performed using a precision colorimeter (WF32, Weifu Optoelectronic Technology Co., Ltd., Shenzhen, China). The color values were expressed as brightness (L*), redness (a*), and yellowness (b*) values. The L* value changed from black (0) to white (100). Furthermore, a* and b* displayed color value changes from greenness (−a*) to redness (+a*) and blueness (−b*) to yellowness (+b*). In addition, the total color difference (∆E) was calculated from the following equation according to Jung [51].
(1)$$\Delta E=\sqrt{{\left(L_{0}-L\right)}^{2}+{\left(a_{0}-a\right)}^{2}+{\left(b_{0}-b\right)}^{2}}$$
where L_0_, a_0_, and b_0_ represented the control values, while L, a, b denoted the values of the treated juice.

#### 3.6.3. Total Phenolic Content

The total phenolic content was determined according to a method delineated by Singleton [52] with some modifications. A calibration curve was prepared using standard gallic acid solutions (0.01~0.08 mg/mL); the total phenolic content in the samples (reported as µg gallic acid equivalent per mL of juice) was calculated from interpolating the respective absorbance values in the calibration curve. The result was expressed as the gallic acid equivalent (mg GAE/mL), and the measurement was repeated three times. Here, 2 mL of *Prunus mume* juice was mixed with 16 mL of a pre-chilled hydrochloric acid (HCl)/methanol (1%, *v*/*v*, 1 mol/L HCl/methanol:distilled water = 1:80:19) mixture and centrifuged at 10,000 rpm for 20 min at 4 °C using a refrigerated centrifuge (5810 R, Eppendorf Company, Hamburg, Germany). Next, 1 mL of clarified juice was diluted to 2 mL with distilled water and mixed with 8 mL of a 7.5% sodium carbonate solution and 4 mL Folin–Ciocalteau reagent diluted six times. The absorbance of the mixture was measured at 765 nm after standing for 1.5 h at room temperature in the dark. The measurement was repeated three times. The total phenol content in *Prunus mume* juice was calculated using a standard linear regression curve (Y = 1.3202X − 0.0344, R^2^ = 0.9993).

#### 3.6.4. Reducing Sugar

The reducing sugar levels were determined via 3,5-dinitrosalicylic acid colorimetry [53]. A calibration curve was prepared using a standard glucose solution (1.0 mg/mL). Respective concentrations of 1.0 mL, 2.0 mL, 3.0 mL, 4.0 mL, 5.0 mL, 6.0 mL, 7.0 mL, and 8.0 mL were placed in 10 mL volumetric flasks with distilled water. After accurately weighing 1.0, 2.0, 3.0, 4.0, 5.0, 6.0, 7.0, and 8.0 mL (1.0 mL/mL) of the prepared solution separately to different 10 mL colorimetric tube, 1.5 mL 3,5-Dinitrosalicylic acid solution was added to each colorimetric tube and held on in boiling water for 5 min, after which rapid cooling and distilled water was fixed to scale. After the absorption value was measured at 540 nm, the standard curve was drawn (Y = 1.1436X − 0.0701, R^2^ = 0.9990).

Sample tests: *Prunus mume* juice was diluted five times with distilled water, 1 ml was measured to determine its absorbance, and the same method used for the standard curve was employed to measure the wavelength. The reducing sugar content in the juice was calculated according to the standard curve without the dilution factor.

#### 3.6.5. 5-Hydroxymethyl Furaldehyde

The 5-hydroxymethylfurfural was determined using a method described by Aguilo [54] with some modifications. Here, 30 mL, 40 mL, 50 mL, 60 mL, 70 mL, 80 mL, and 90 mL of the 5-hydroxymethylfurfural standard solution were respectively placed in 10 mL volumetric flasks with distilled water. The absorbance was measured at 433 nm to obtain a standard curve (Y = 0.1262X − 0.1911, R^2^ = 0.9991).

A 6 mL sample was added to the test tube, followed by a 4 mL trichloroacetic acid solution (g/L) and 4 mL 2-thiobarbituric acid (0.025 mol/L) at 40 °C water for 50 min. The mixtures were cooled rapidly, and the absorbance was measured at 443 nm, after which the results were calculated according to the standard curve.

#### 3.6.6. Amino Acid Nitrogen

An appropriate amount of the sample solution (1 mg- to 5 mg of amino nitrogen) was added to a beaker with five drops of 30% hydrogen peroxide. The organic acids in the sample were gradually neutralized with a 0.1 mol/L sodium hydroxide standard solution using an electromagnetic stirrer. When the pH reached 7.5, the 0.05 mol/L sodium hydroxide standard titration solution was adjusted to pH 8.1 and remained unchanged for 1 min, after which a 10 mL~15 mL neutral formaldehyde solution was gradually added. After 1 min, the 0.05 mol/L sodium hydroxide standard solution was titrated to pH 8.1, and the ml consumption was recorded.

#### 3.6.7. Determination of the pH and Total Soluble Solid

A total of 25 mL *Prunus mume* juice was placed in a beaker, and the pH value was recorded after the reading was stable. Each sample was measured three times, and the results were averaged.

An Abbe refractometer (A610, Jinan Haieng Instrument Co., Ltd., Shandong, China) was zeroed with distilled water, after which 200 uL *Prunus mume* juice was absorbed on the surface of the glass mirror hole, yielding a reading of 25 ± 2 °C at room temperature. The results were calculated using °Brix.

### 3.7. Microbiological Examination

The microbiological counts were determined using a method described by Chen [55] with some modifications. Here, 25 mL of the sample was mixed with a 225 mL 0.85% sodium chloride solution, homogenized, and used to prepare 10^−1^ dilutions. The aerobic bacterial count was determined via the pour plate method, using plate count agar (Beijing Aobo Star Biotechnology Co., Ltd., Beijing, China) as a medium. The plates were incubated at 37 °C for 48 h ± 2 h. Rose- Bengal medium was used to determine the total mold-yeast counts. The plates were incubated at 28 °C for 72 h.

### 3.8. Storage Treatment of Prunus mume Juice

The *Prunus mume* juice samples were treated with P and HHP sterilization, and then stored at 4 °C, 25 °C, and 35 °C (temperature fluctuation range ± 1.0 °C, relative humidity 75%) for 15 days in dark. The CK group was used as the control. The samples were sampled and determined at 0, 3, 6, 9, 12, and 15 days, respectively. The 15th day was the test end point of the storage test.

### 3.9. Statistical Analysis

Each group of experiments was repeated three times, and the experimental results are expressed as mean ± SD, while the data were analyzed using SPSS 23.0. The PCA and LDA graphs of the E-nose were obtained via WinMuster software, while those presenting the other experimental results were drawn using Origin 8.5.

## 4. Conclusions

This study compared the impact of different sterilization methods, such as P, O_3_, US, and HHP, on the physicochemical properties, microbial indexes, and changes in physicochemical properties and microorganisms of *Prunus mume* juice during storage to determine the most suitable processing technique. The results indicate P treatment reduces the non-enzymatic browning degree while significantly affecting (*p* < 0.05) the total phenols, TSS, and pH; however, reducing sugar, 5-HMF, and amino acid nitrogen content did not exhibit a significant change (*p* > 0.05); the sterilization effect of P treatment was significant; sterilization rate was 100%. O_3_ treatment substantially affected the browning degree, ΔE, total phenols, reducing sugar, amino acid nitrogen, TSS and pH value in the *Prunus mume* juice (*p* < 0.05), the sterilization effect of O_3_ treatment was general, and the sterilization rate was 70%. US treatment had a considerable impact on the degree of browning, ΔE, reducing sugar, 5-HMF, TSS, and pH value (*p* < 0.05); the sterilization effect on the juice was general; and the sterilization rate was 80%. HHP treatment decreased the degree of non-enzymatic browning of the *Prunus mume* juice and significantly affected ΔE, TSS, and pH (*p* < 0.05); while the sterilization rate was as high as 90%. During storage at 4 °C, 25 °C and 37 °C, the browning degree and ΔE value of *Prunus mume* juice increased significantly, and the contents of Maillard reaction substrates such as total phenols, reducing sugar and amino acid nitrogen decreased significantly. The content of 5-HMF fluctuated and increased in the dynamic reaction system of *Prunus mume* juice. The TSS and pH values of *Prunus mume* juice were relatively stable and did not change significantly. Under the same sterilization condition, the transverse comparison of storage temperature showed that P and HHP treatments had better quality at 4 °C, and that low temperature storage was beneficial for maintaining the quality of *Prunus mume* juice. Therefore, P treatment or HHP treatment combined with low temperature storage could achieve more ideal storage effect. Therefore, P and HHP technology are a more suitable for *Prunus mume* juice sterilization, minimally affecting its overall quality and better preserving its color, and physicochemical components.

## Figures and Tables

**Figure 1 molecules-27-01197-f001:**
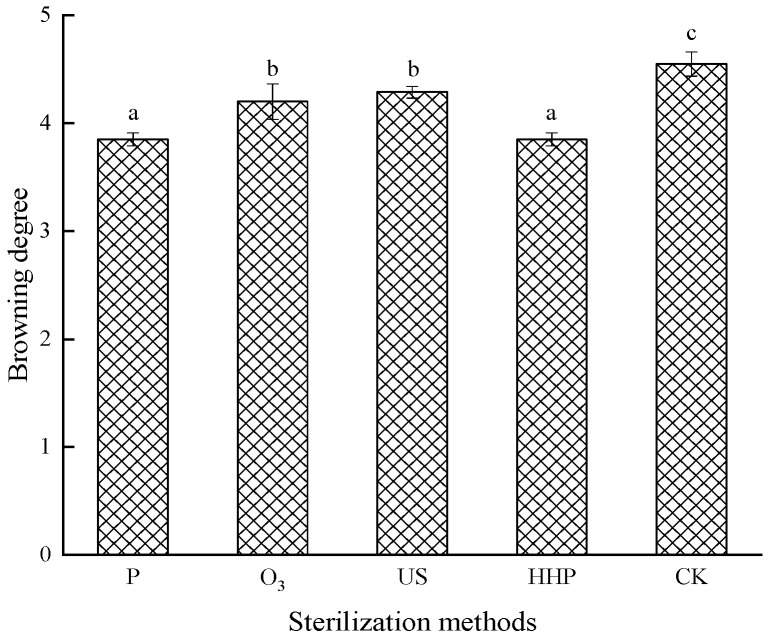
The effect of pasteurization (P), ozone (O_3_), ultrasonic (US), and high-hydrostatic-pressure (HHP) sterilization methods and non-sterilization (control check, CK) on the browning degree of *Prunus mume* juice. Values with the different letters in the column are significantly different by Duncan’s multiple range test (*p* < 0.05).

**Figure 2 molecules-27-01197-f002:**
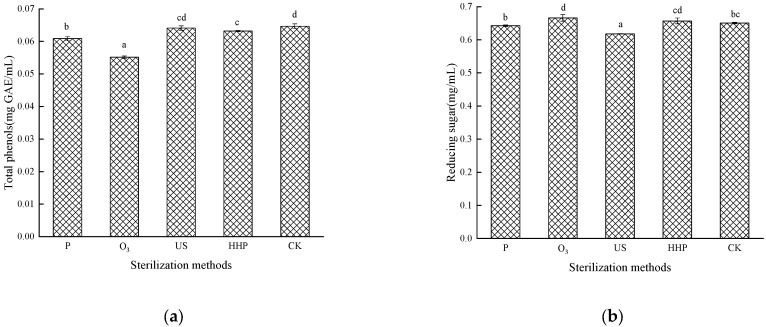
The effect of pasteurization (P), ozone (O_3_), ultrasonic (US), and high-hydrostatic-pressure (HHP) sterilization methods and non-sterilization (control check, CK) on total phenolic content (**a**) and (**b**) the reducing sugar content in the *Prunus mume* juice. Values with the different letters in the column are significantly different by Duncan’s multiple range test (*p* < 0.05).

**Figure 3 molecules-27-01197-f003:**
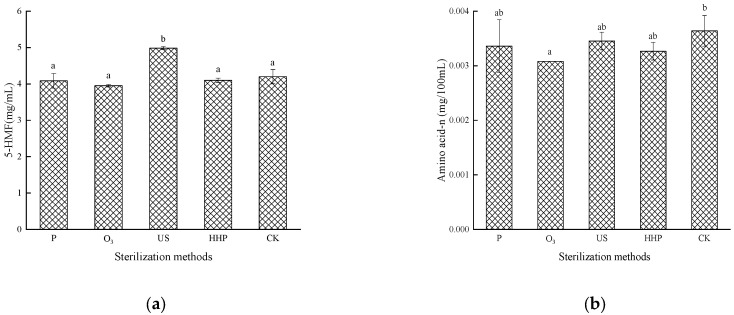
The effect of pasteurization (P), ozone (O_3_), ultrasonic (US), and high-hydrostatic-pressure (HHP) sterilization methods and non-sterilization (control check, CK) on the 5-hydroxymethyl furaldehyde levels (**a**) and the amino acid- nitrogen levels (**b**) in the *Prunus mume* juice. Values with the different letters in the column are significantly different by Duncan’s multiple range test (*p* < 0.05).

**Figure 4 molecules-27-01197-f004:**
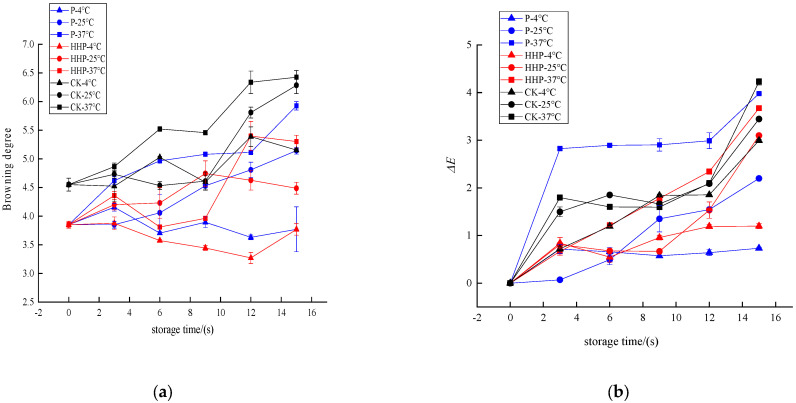
The effect of pasteurization (P) and high-hydrostatic-pressure (HHP) sterilization methods and non-sterilization (control check, CK) on browning degree (**a**) and color measurement (**b**) in the *Prunus mume* juice.

**Figure 5 molecules-27-01197-f005:**
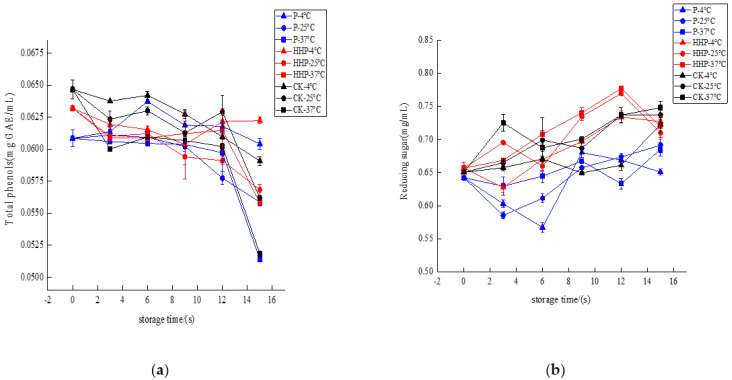
The effect of pasteurization (P) and high-hydrostatic-pressure (HHP) sterilization methods and non-sterilization (control check, CK) on total phenol content (**a**) and reducing sugar (**b**) in the *Prunus mume* juice.

**Figure 6 molecules-27-01197-f006:**
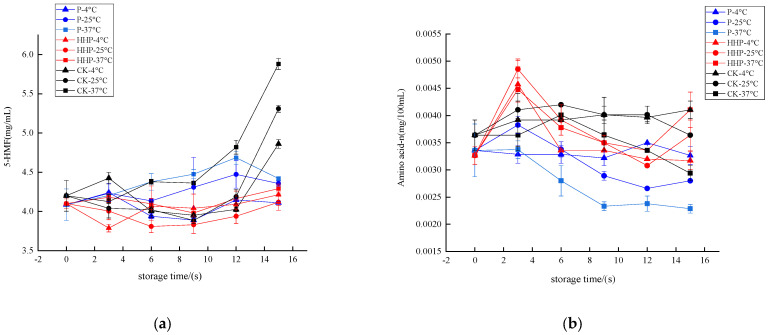
The effect of pasteurization (P) and high-hydrostatic-pressure (HHP) sterilization methods and non-sterilization (control check, CK) on the 5-HMF levels (**a**) and the amino acid nitrogen content (**b**) in the *Prunus mume* juice.

**Figure 7 molecules-27-01197-f007:**
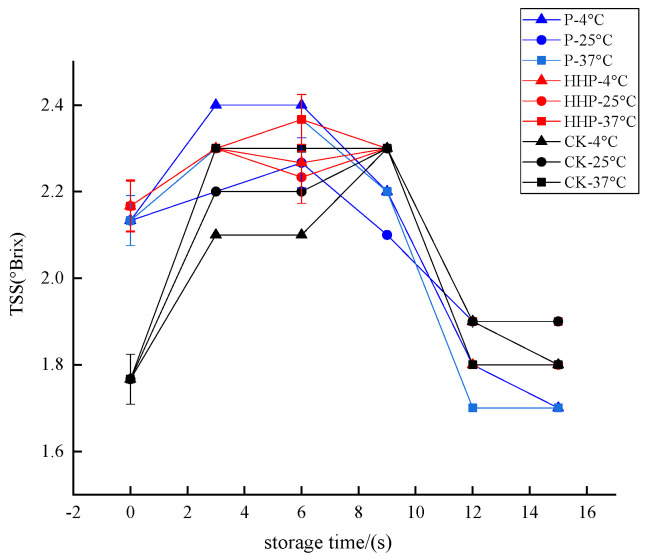
The effect of pasteurization (P) and high-hydrostatic-pressure (HHP) sterilization methods and non-sterilization (control check, CK) on the TSS in the *Prunus mume* juice.

**Table 1 molecules-27-01197-t001:** The effect of pasteurization (P), ozone (O_3_), ultrasonic (US), and high-hydrostatic-pressure (HHP) sterilization methods and non-sterilization (control check, CK) on the ΔE of the *Prunus mume* juice.

Sterilization Methods	L*	a*	b*	ΔE
P	13.14 ± 0.03 ^a^	1.06 ± 0.03 ^a^	1.57 ± 0.03 ^a^	1.99 ± 0.01 ^d^
O_3_	13.64 ± 0.15 ^b^	2.15 ± 0.04 ^c^	1.73 ± 0.03 ^b^	1.74 ± 0.06 ^c^
US	14.64 ± 0.09 ^d^	1.11 ± 0.02 ^a^	1.53 ± 0.05 ^a^	1.24 ± 0.04 ^b^
HHP	13.82 ± 0.08 ^c^	1.06 ± 0.05 ^a^	2.25 ± 0.03 ^c^	1.05 ± 0.05 ^a^
CK	14.72 ± 0.04 ^d^	1.25 ± 0.01 ^b^	2.76 ± 0.06 ^d^	-

Note: The data are expressed as mean ± SD (*n* = 3). Different superscript letters in the same column. denote significant differences. Values with the different letters in the column are significantly different by Duncan’s multiple range test (*p* < 0.05).

**Table 2 molecules-27-01197-t002:** The effect of pasteurization (P), ozone (O_3_), ultrasonic (US), and high-hydrostatic-pressure (HHP) sterilization methods and non-sterilization (control check, CK) on the total soluble solid and pH of the *Prunus mume* juice.

Sterilization Methods	TSS (°Brix)	pH
P	2.13 ± 0.06 ^c^	2.34 ± 0.01 ^c^
O_3_	2.03 ± 0.06 ^b^	2.32 ± 0.02 ^ab^
US	2.20 ± 0.00 ^c^	2.32 ± 0.06 ^b^
HHP	2.17 ± 0.06 ^c^	2.31 ± 0.06 ^ab^
CK	1.77 ± 0.06 ^a^	2.30 ± 0.00 ^a^

Note: The data are expressed as mean ± SD (*n* = 3). Different superscript letters in the same column denote significant differences. Values with the different letters in the column are significantly different by Duncan’s multiple range test (*p* < 0.05).

**Table 3 molecules-27-01197-t003:** The effect of pasteurization (P), ozone (O_3_), ultrasonic (US), and high-hydrostatic-pressure (HHP) sterilization methods and non-sterilization (control check, CK) on the microbial counts in the *Prunus mume* juice.

Sterilization Methods	APC (CFU/mL)	M&Y (CFU/mL)
P	ND	ND
O_3_	15	ND
US	10	ND
HHP	5	ND
CK	50	15

Note: The data are expressed as mean ± SD. ND = not detected.

**Table 4 molecules-27-01197-t004:** The effect of pasteurization (P) and high-hydrostatic-pressure (HHP) sterilization methods and non-sterilization (control check, CK) on pH of *Prunus mume* juice during storage.

Storage Temperature (°C)	Storage Time (d)	pH		
		P	HHP	CK
4	0	2.34 ± 0.01 ^a^	2.31 ± 0.01 ^a^	2.30 ± 0.00 ^a^
	3	2.30 ± 0.00 ^a^	2.28 ± 0.00 ^a^	2.27 ± 0.00 ^a^
	6	2.29 ± 0.01 ^a^	2.29 ± 0.01 ^a^	2.29 ± 0.01 ^a^
	9	2.30 ± 0.01 ^a^	2.28 ± 0.01 ^a^	2.27 ± 0.00 ^a^
	12	2.31 ± 0.01 ^a^	2.29 ± 0.01 ^a^	2.29 ± 0.01 ^a^
	15	2.31 ± 0.01 ^a^	2.29 ± 0.01 ^a^	2.28 ± 0.01 ^a^
25	0	2.34 ± 0.01 ^a^	2.31 ± 0.01 ^a^	2.30 ± 0.00 ^a^
	3	2.27 ± 0.00 ^a^	2.26 ± 0.00 ^a^	2.26 ± 0.00 ^a^
	6	2.29 ± 0.01 ^a^	2.27 ± 0.01 ^a^	2.28 ± 0.01 ^a^
	9	2.28 ± 0.01 ^a^	2.28 ± 0.01 ^a^	2.27 ± 0.01 ^a^
	12	2.32 ± 0.01 ^a^	2.29 ± 0.01 ^a^	2.30 ± 0.01 ^a^
	15	2.29 ± 0.01 ^a^	2.28 ± 0.01 ^a^	2.29 ± 0.01 ^a^
37	0	2.34 ± 0.01 ^a^	2.31 ± 0.01 ^a^	2.31 ± 0.00 ^a^
	3	2.30 ± 0.00 ^a^	2.25 ± 0.00 ^a^	2.25 ± 0.00 ^a^
	6	2.30 ± 0.01 ^a^	2.31 ± 0.00 ^a^	2.30 ± 0.01 ^a^
	9	2.30 ± 0.01 ^a^	2.29 ± 0.00 ^a^	2.26 ± 0.01 ^a^
	12	2.31 ± 0.01 ^a^	2.31 ± 0.00 ^a^	2.30 ± 0.01 ^a^
	15	2.31 ± 0.01 ^a^	2.29 ± 0.01 ^a^	2.29 ± 0.01 ^a^

Note: The data results are expressed as Mean ± SD (*n* = 3); there were significant differences in the same column with different English letters, values with the different letters in the column are significantly different by Duncan’s multiple range test (*p* < 0.05).

**Table 5 molecules-27-01197-t005:** The effect of pasteurization (P) and high-hydrostatic-pressure (HHP) sterilization methods and non-sterilization (control check, CK) on microbial counts of *Prunus mume* juice during storage.

Determination Standard	Storage Time/d	Storage Temperature/°C								
		4			25			37		
		P	HHP	CK	P	HHP	CK	P	HHP	CK
APC(CFU/mL)	0	ND	5	50	ND	5	50	ND	5	50
	3	ND	<1	ND	ND	5	ND	ND	5	ND
	6	ND	<1	5	ND	<1	10	ND	5	ND
	9	ND	ND	ND	ND	ND	91	ND	ND	ND
	12	ND	ND	1	ND	ND	-	ND	ND	-
	15	ND	ND	-	ND	ND	-	ND	ND	-
M&Y(CFU/mL)	0	ND	ND	15	ND	ND	15	ND	ND	15
	3	ND	ND	ND	ND	ND	5	ND	ND	<1
	6	ND	ND	ND	ND	ND	5	ND	ND	<1
	9	ND	ND	5	ND	ND	ND	ND	ND	ND
	12	ND	ND	ND	ND	ND	ND	ND	ND	ND
	15	ND	ND	ND	ND	ND	ND	ND	ND	ND

Note: Data results are expressed as Mean ± SD; ND = not detected; - colonies spread.

## Data Availability

Not applicable.

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
