# Peer review of "Effect of Different Sterilization Methods on the Microbial and Physicochemical Changes in Prunus mume Juice during Storage"

_molecules, 2022, doi:10.3390/molecules27041197_

Round 1

Reviewer 1 Report

Dear Author

I found the MS valuable for using novel technologies in extending the shelf life and quality indices of fruit juices. However there are points which must be clarified

1- The introduction is not well defined. The idea propagation is missed.

2- Each method and preparation procedure must be used from distinct references.

3- Describe the referactometer precision. How could you obtain the TSS values with 2 decimals for example 2.03±0.06.

4- Discussion about effect of methods on total count is poor.

Author Response

Dear reviewers:       

We have tried our best to revise and improve the manuscript and made changes in the manuscript according to the Reviwers’ good comments. And here we did not list the changes but marked in red in revised paper.  

We appreciate for Editors/Reviewers’ warm work earnestly, and hope that the corrections will meet with approval.

Once again, thank you very much for your comments and suggestions.

We look forward to your information about my revised papers and thank you for your good comments.    

Yours sincerely,

Yuan Ma

Reviewer 2 Report

The objective of this study was to compare the effect of pasteurization (P), as well as O3, Ultrasonic, and high hydrostatic pasterilization, on the physicochemical properties, especially the browning factors and flavor of Prunus mume juice. The study considered also the impact of these techniques on the microbiological quality and physic-chemical properties of the prepared and stored juices.  This comparative study has a scientific and technological significance.

Generally the English should be revised and edited for more simplicity and clarity.

Introduction

Revise and reformulate the first sentence (L29-30) in the introduction to be clearer.

The last sentence of (L34-36) of the first paragraph in the introduction needs to be clearer. It is referring to water content and acidity without clarifying the connection between the two.

L40 (during sterilization and the later storage process) change into (during sterilization and the subsequent storage process).

L42 (easily leading to browning) change into (quickly leading to browning).

Conclusion

The sentence (The sterilization effect is significant at a sterilization rate of 100%.) is not clear. Please, elucidate.

Last sentence in the conclusion may better be modified into (Therefore, P and HHP technology are more suitable for Prunus mume juice sterilization, being minimally affecting its overall quality and better preserving its color, flavor, and physicochemical components.)

22.11.2021

L47 (Therefore, this work examines the effect of pasteurization (P), as well as O3, US, and HHP sterilization)

L49-50, (the impact of microorganisms) change into (the impact on microorganisms)

L50 change examines into compares

Change (number of bacterial colony) to (number of the bacterial colony).

Materials and Methods

The second sentence under 2.1. (L58-60) is not clear and should be modified. It can for example divided into two small sentence as follows: (The Prunus mume kernels were removed. The, one volume of the pulp was mixed with three volumes of water and squeezed using a juice extractor).

L62, (Next, 0.06% of a pectic enzyme). Describe briefly this enzyme. What is its scientific name and specific activity.

L63-66, reformulate this sentence in clearer way. You can start with (A mixture of …., ….. and ….. was prepared and added to the clarified juice.

L66 (added to the clarified juice in optimized proportion.). Define what is this optimized proportion? Or explain how the proportion was optimized.

L74-80 (Section 2.3). Please explain how ozone was applied.

L94 (Here, 6 mL of juice was mixed with an equal quantity). Delete the word (Here). The sentence can start with: A volume of juice (6 ml) was mixed with an equal quantity)ز

The section should be simplified and reduced in volume.

L187-189, the sentence (Each Prunus mume juice was put into a glass beaker (40 mL) sealed with plastic cap for 30 mins to ensure the volatile compounds which released from the Prunus mume juice to fill the beaker and to get equilibrium) should be corrected into (A volume (.....mL) of  Prunus mume juice was placed in a glass beaker (40 mL) sealed with a plastic cap for 30 min to ensure the fully equilibrated release of the volatile compounds. ). You should mention the volume of the juice.

Results

This part should be corrected to (3.Results and Discussion). The discussion should be further consolidated. The differences between the sterilization techniques should be discussed more deeply.

Under each figure explain the definition of each abbreviation. Every figure should be self-explained.

Figure 1, mention the units of measuring the browning degree, under the table and in the text.

Under all figures, CK is presented as one of the sterilization methods, but it is not explained anywhere what does CK refer to, either under the figure or in the relevant text. The discussion of this treatment is never referred to. Is it the non-sterilized treatment? In either case it should be defined under the figure and in the text.

Author Response

(The authors gave the same response as above.)

Round 2

Reviewer 2 Report

The authors have largely improved their text and presentation.

Author Response

Dear Editors:       

We have tried our best to revise and improve the manuscript and made changes in the manuscript according to the Reviwers’ good comments. And here we did not list the changes but marked in red in revised paper.  

We appreciate for Editors/Reviewers’ warm work earnestly, and hope that the corrections will meet with approval.

Once again, thank you very much for your comments and suggestions.

We look forward to your information about my revised papers and thank you for your good comments.    

Yours sincerely,

Yuan Ma
